Measurement of sedentary behaviour in population health surveys: a review and recommendations

Prince Stephanie A. sprinceware@ottawaheart.ca 1 4
LeBlanc Allana G. 1
Colley Rachel C. 2
Saunders Travis J. 3
1 Division of Prevention and Rehabilitation, University of Ottawa Heart Institute , Ottawa , Ontario , Canada
2 Health Analysis Division, Statistics Canada , Ottawa , Ontario , Canada
3 Department of Applied Human Sciences, University of Prince Edward Island , Charlottetown , Prince Edward Island , Canada
4 Current affiliation:  Centre for Surveillance and Applied Research, Public Health Agency of Canada
Gill Jason
Electronic publication date: 2017 Dec 11
Publication date: 2017
Volume: 5
Electronic Location ID: e4130
Received 2017 Jun 26; Accepted 2017 Nov 14
Copyright: ©2017 Prince et al.
Copyright year: 2017
Copyright holder: Prince et al.
License: This is an open access article distributed under the terms of the Creative Commons Attribution License, which permits unrestricted use, distribution, reproduction and adaptation in any medium and for any purpose provided that it is properly attributed. For attribution, the original author(s), title, publication source (PeerJ) and either DOI or URL of the article must be cited.
License URL: https://creativecommons.org/licenses/by/4.0/

Keywords: Validity, Survey, Questionnaire, Sedentary behaviour, Reliability

Funding: Public Health Agency of Canada Jeanne and J.-Louis Lévesque Research Professorship in Nutrisciences and Health This work was supported by the Public Health Agency of Canada. Travis Saunders is supported by the Jeanne and J.-Louis Lévesque Research Professorship in Nutrisciences and Health. The funders had no role in study design, data collection and analysis, decision to publish, or preparation of the manuscript.

==============================
Background

The purpose of this review was to determine the most valid and reliable questions for targeting key modes of sedentary behaviour (SB) in a broad range of national and international health surveillance surveys. This was done by reviewing the SB modules currently used in population health surveys, as well as examining SB questionnaires that have performed well in psychometric testing.

Methods

Health surveillance surveys were identified via scoping review and contact with experts in the field. Previous systematic reviews provided psychometric information on pediatric questionnaires. A comprehensive search of four bibliographic databases was used to identify studies reporting psychometric information for adult questionnaires. Only surveys/studies published/used in English or French were included.

Results

The review identified a total of 16 pediatric and 18 adult national/international surveys assessing SB, few of which have undergone psychometric testing. Fourteen pediatric and 35 adult questionnaires with psychometric information were included. While reliability was generally good to excellent for questions targeting key modes of SB, validity was poor to moderate, and reported much less frequently. The most valid and reliable questions targeting specific modes of SB were combined to create a single questionnaire targeting key modes of SB.

Discussion

Our results highlight the importance of including SB questions in survey modules that are adaptable, able to assess various modes of SB, and that exhibit adequate reliability and validity. Future research could investigate the psychometric properties of the module we have proposed in this paper, as well as other questionnaires currently used in national and international population health surveys.

Introduction

Sedentary behaviour (SB; sitting, and activities that require very low energy expenditure and done while sitting, reclining or lying down (Tremblay et al., 2017)) is a unique risk factor for several chronic diseases and mortality (Biswas et al., 2015; Wilmot et al., 2012; Thorp et al., 2011; Carson et al., 2016; Colley et al., 2011a). Recognition and interest in this risk factor has prompted the inclusion of measures of SB in population health surveillance surveys around the world (Colley et al., 2011a; Colley et al., 2011b; Matthews et al., 2008; Australian Bureau of Statistics, 2011; Scottish Government, 2015). While self-report tools provide information about mode and domains of SB, little is known about their validity (the degree to which the questionnaire measures what it claims to measure) and reliability (the degree to which a questionnaire can produce consistent and reproducible results) (Carson et al., 2016; Atkin et al., 2012). Habitual patterns of SB can be measured objectively using accelerometers and inclinometers, but these methods are often too time or resource intensive for inclusion in population-level health surveys and studies. Further, these objective methodologies are unable to distinguish between different domains (e.g., occupational/school, transportation, leisure, domestic) and modes (e.g., TV, computer use, reading, car driving) of SB. This is an important issue, given that some modes of SB appear to be more consistently associated with indicators of poor health than others. For example, the relationship between total SB and health outcomes is often weaker than for some specific modes of SB, especially TV viewing and total screen time (Carson et al., 2016; Ford & Caspersen, 2012; Ekelund et al., 2016). A smaller body of research suggests that sedentary transportation may also show deleterious associations with health (Sugiyama et al., 2016), whereas reading has been shown to be benign or even beneficial (Carson et al., 2016; Bavishi, Slade & Levy, 2016). It is important to note, however, that further research is still needed to identify whether these associations are independent of other confounding factors such as food consumption and socio-economic status.

While two recent systematic reviews have examined the reliability and validity of SB questionnaires in pediatric populations (Hidding et al., 2016; Lubans et al., 2011), no reviews have compared the psychometric properties of SB questionnaires in adults, and none have examined those used in population level surveys. Therefore, the objectives of the present review were to: (1) summarize the available self-report tools for assessing the most common modes of SB including TV viewing, computer use, total screen time, reading, sedentary transportation, and total SB in national and international population surveillance surveys; and, (2) to identify the most valid and reliable questions/questionnaires for assessing total and individual modalities of SB. We aim to provide readers with practical and evidence-informed information to support the development of future population health surveys.

Methods

Inclusion and exclusion criteria

The present review focuses on questionnaires used in national and international surveys, as well as those that have undergone formal testing for validity and/or reliability. Activity diaries and ecological momentary assessment tools were excluded from the review due to their low level of practicality within the context of population health surveys. Surveys and any associated validity/reliability testing had to be in English or French to be included in this review.

National/international survey questions

To be included in the present review, surveys had to assess SB (e.g., sitting/reclining/lying and an energy expenditure ≤ 1.5 metabolic equivalents (Tremblay et al., 2017)), as opposed to the lack of physical activity (often referred to as physical inactivity). Questionnaires were excluded if we were unable to obtain complete wording for SB items within the questionnaire. Questionnaires used to assess SB in multiple regions in an individual country were considered national in scope, while those that assessed SB in multiple countries were considered international. Surveys that examined only a specific location or region within a country were excluded, as were surveys that examined special populations (e.g., those with a specific disease or condition).

Studies evaluating questionnaire reliability and validity

To be included in the present review, individual studies required at least 30 participants per analysis to ensure adequate power (80%, α = 0.05) to identify a moderate correlation (r = 0.50) between self-report and objective measures.

Search strategy

National/international survey questionnaires

National and international survey questionnaires were identified via the reference databases of the authors and through a scoping review using the Google search engine. An email was also sent to members of the Sedentary Behaviour Research Network (SBRN; a research network of over 1,100 scientists with an interest in SB, http://www.sedentarybehaviour.org asking for help in the identification of additional national and international surveys with questions or components measuring SBs.

Studies evaluating questionnaire reliability and validity

Similar to the search for national and international SB questionnaires, studies examining the validity and reliability of SB questionnaires were first identified via personal reference databases, then through email correspondence with SBRN members. During this process, we identified two recent systematic reviews that had summarized the reliability and validity of SB questionnaires in children and youth (Hidding et al., 2016; Lubans et al., 2011). These reviews provided a high quality summary of the current evidence and were used to inform our discussion on reliability and validity of SB questionnaires among the pediatric population.

We were unable to identify any similar review of SB questionnaires among adults. As a result, we performed a search of the literature to identify relevant studies in adults (aged >18 years). A search strategy (Table S1) was carried out in four electronic databases including: Ovid MEDLINE(R) In-Process (1946—November Week 1 2016); Ovid PsycINFO (1806 to November Week 1 2016); EBSCOhost SPORTDiscus (1830 to November 2016); and EBM Reviews—Cochrane Database of Systematic Reviews 2005 to November 1, 2016). The search sought to identify studies that reported on the validity and/or reliability of a self-report tool (i.e., questionnaire, survey) that measures SB.

Assessment of reliability and validity

In the context of this review, a SB measurement tool with high reliability consistently provides similar estimates of SB across multiple trials. Test-retest reliability is often assessed in SB research using an intraclass correlation coefficient (ICC). Cronbach’s α is used to test for internal consistency of a tool. Both measures produce values ranging from 0 to 1; where 1 represents perfect reliability and consistent results and 0 represents no reliability or inconsistent results. It is therefore ideal to have an ICC and Cronbach α as close to 1 as possible, with anything over 0.75 considered excellent. In the present review, an ICC between 0.60 and 0.74 was considered good, an ICC between 0.40 and 0.59 was considered fair, and an ICC < 0.40 was considered poor (Cicchetti, 1994).

Identifying whether a self-report tool is able to accurately quantify SB is referred to as criterion validity. Validity of a self-report SB measure is often assessed against objective measures (e.g., activPAL™, accelerometer, direct observation). The majority of validation studies report a level of correlation between two measures (e.g., questionnaire and accelerometer-measured sedentary time) and similar to the ICC, a correlation coefficient closer to 1 was used to indicate a stronger relationship. We also examined, when available, mean differences and levels of agreement between the self-report and objective measures.

Validity and reliability statistics were extracted in the format provided by the individual studies. Inclinometry was considered the gold standard for total SB and sitting time (e.g., activPAL), which has been shown to have the highest sensitivity for distinguishing between sitting and standing (Aminian & Hinckson, 2012; Dowd, Harrington & Donnelly, 2012). Accelerometry was also considered as a criterion measure to assess validity. Although accelerometry provides an objective measurement of time spent sedentary, it is not as sensitive as inclinometry for measuring SB because of its inability to distinguish between stationary standing and sitting and may therefore misclassify some standing time as SB (Aminian & Hinckson, 2012; Dowd, Harrington & Donnelly, 2012). Inclinometry and accelerometry were not considered appropriate criterion standards for specific SB modalities (e.g., TV time, reading). Rather, direct observation or detailed diaries/logs were considered as useful measures for looking at the validity of questionnaires which measured specific modalities of SB.

Results

Sedentary behaviour questions used in national/international surveys and studies

The review identified a total of 16 pediatric and 18 adult national/international surveys and large national epidemiological studies assessing at least one modality of SB (Table S1). Pediatric surveys meeting inclusion criteria were used in 38 countries, while we identified adult surveys used in 22 countries. Surveys included as few as one question (e.g., Global Physical Activity Questionnaire (GPAQ), European Prospective Investigation into Cancer and Nutrition-Potsdam Study (EPIC)), and as many as 12 questions (Children’s Leisure Activities Survey (CLASS)) related to SB. Although all included questionnaires employed English or French versions, many had also been translated into other languages for specific populations. There was a considerable lack of published literature reporting on psychometric testing for the majority (pediatric = 63%, adult = 56%) of the questionnaires used in national and international surveys for all age groups. SB modalities varied across questionnaires, with TV viewing time being the most frequently assessed (pediatric = 100%, adults = 72%). Computer and/or video game time were also frequently assessed, especially among pediatric populations (pediatric = 88%, adults = 39%). Fewer surveys included questions related to reading (pediatric = 44%, adults = 50%) or sedentary transport (pediatric = 13%, adult = 33%). The wording of questions varied across surveys, although often in relatively trivial ways (e.g., reporting in hours versus minutes). Many (pediatric = 38%, adults = 44%), but not all, of the surveys referred to a specific time period for recall (e.g., the past week, four weeks, three months, or year), and reminded participants to focus on a “typical” or “usual” day or week in that time period. Some surveys focused on hours per day of each SB mode, whereas others focused on hours per week. Some surveys asked about an average of the days of the week, while others had separate questions for school/work/week days and weekends. Several (pediatric = 63%, adults = 28%) of the surveys separated their questions for each modality of SB (e.g., Canadian Health Measures Survey (CHMS), ISCOLE, HBSC, etc). Some surveys employed the use of a grid or list of SB modalities (e.g., COMPASS) and required participants to enter daily time for each SB.

Reliability and validity of individual questionnaires

The reliability and validity of individual questionnaires has been summarized in Table S3. Items/questions from these questionnaires can be found on the SBRN website http://www.sedentarybehaviour.org/sedentary-behaviour-questionnaires/). We identified 14 questionnaires from previous systematic reviews (Hidding et al., 2016; Lubans et al., 2011) which have undergone psychometric testing in a pediatric population. Through our search strategy (Fig. 1), we identified 35 adult questionnaires with published psychometric properties (multiple papers reported on the psychometric testing of the same questionnaire) that examined the validity and/or reliability of adult SB questionnaires. Included questionnaires contained as few as one question (e.g., GPAQ, Yale Physical Activity Survey for Older Adults (YPAS), Past Year Physical Activity Questionnaire, Past-Week Modifiable Activity Questionnaire (PWMAP), Modified MONICA Optional Study on Physical Activity Questionnaire (MOSPA-Q)) (Cleland et al., 2014; Herrmann et al., 2013; Chau et al., 2012; Pettee Gabriel et al., 2011; Aguilar-Farias et al., 2015; Gennuso, Matthews & Colbert, 2015), and as many as 23 (Adolescent Sedentary Activities Questionnaire (ASAQ) (Hardy, Booth & Okely, 2007)) items related to SB. Although we identified studies examining the reliability of questions related to key SB modalities, only the measurement of total SB and total sitting time employed appropriate criterion standards for validity.

Figure 1 Flow diagram of literature search for adult questionnaires.

SB, sedentary behaviour; SB, Sedentary Behaviour Research Network.

TV viewing

Among preschool-aged children and youth, both the Preschool-aged Physical Activity Questionnaire (Pre-PAQ) (ICC = 0.70–0.88, 95% CI: not reported (NR)) (Dwyer et al., 2011) and the proxy-report questionnaire used in the Health, Eating and Play Study (HEAPS) (ICC = 0.78, 95% CI [0.69–0.84]) (Salmon, Campbell & Crawford, 2006), had excellent levels of reliability. Testing of the COMPASS questionnaire in children in grades 9-12 yielded a fair ICC of 0.56 (95% CI: NR), and a Cronbach’s α of 0.74 (Leatherdale, Laxer & Faulkner, 2014), which was the highest identified in this age group. Among adults, the Sedentary Behavior Questionnaire (SBQ) demonstrated excellent reliability for weekday (ICC = 0.86, 95% CI [0.76–0.92]) and for weekend (ICC = 0.83, 95% CI [0.72–0.90]) TV viewing (Rosenberg et al., 2010), while the Past Week Modifiable Activity Questionnaire (PWMAQ; ICC = 0.67, 95% CI [0.61–0.71]) (Pettee Gabriel et al., 2011), Salmon SB questionnaire (ICC = 0.82, 95% CI: [0.75–0.87]) (Salmon et al., 2003)/Measure of Older Adults Sedentary Time (MOST; ICC = 0.76, 95% CI [0.62–0.86]) (Salmon et al., 2003; Gardiner et al., 2011), Sedentary, Transportation and Activity Questionnaire (STAQ; ICC = 0.79, 95% CI [0.61–0.89]) (Mensah et al., 2016), and the SIT-Q (ICC = 0.84, 95% CI [0.75–0.90]) (Lynch et al., 2014) also had very reliable questions for TV time. These questionnaires varied in both the wording of the questions and in response categories, suggesting that a variety of approaches provide reliable results for TV viewing. Few studies have compared appropriate objective measures of TV-specific SB to self-reported TV time. Among children, the Youth Risk Behaviour Survey TV time questions were validated against a 7-day TV log and exhibited a moderate correlation (r = 0.46) (Schmitz et al., 2004). Among adults, the Salmon SB questionnaire was poorly correlated (r = 0.3, p < 0.01) with 3-day logs for measures of self-reported TV time (Salmon et al., 2003).

Computer, tablet and video game use

Compared to TV viewing, relatively few (pediatric = 19%, adult = 11%) questionnaires have undergone psychometric testing for items related to computer use. Among pre-school aged children and youth, the Pre-PAQ proxy-report questionnaire demonstrated high levels of test-retest reliability for computer and video game playing (ICC = 0.82–0.85, 95% CI: NR) (Dwyer et al., 2011). The COMPASS questionnaire had slightly lower, but still good levels of reliability on questions related to computer and video game use (ICC = 0.65, 95% CI: NR, Cronbach’s α = 0.79) and surfing the internet (ICC = 0.71, 95% CI: NR, Cronbach’s α = 0.84) among high school students (Leatherdale, Laxer & Faulkner, 2014). Among adults, the Gennuso et al. SB questionnaire (ICC = 0.93, p < .001) (Gennuso et al., 2016) and the Measure of Older Adults’ Sedentary time (MOST) (ICC = 0.79, 95% CI [0.65–0.86]) (Gardiner et al., 2011) had very high reliability for the question targeting computer and internet use. Similarly, the SBQ (Rosenberg et al., 2010) has shown high reliability (weekday: ICC = 0.83; 95% CI [0.72–0.90], weekend: ICC = 0.80; 95% CI [0.67–0.88]) for a question focusing on computer and video game use. The Marshall Sitting Time Questionnaire asks a single question targeting home-based computer use and has demonstrated good reliability (women: weekday ICC = 0.63, 95% CI [0.52–0.71]; weekend ICC = 0.72, 95% CI [0.64–0.79], men: weekday ICC = 0.62, 95% CI [0.48–0.73]; weekend ICC = 0.59, 95% CI [0.44–0.71]) (Marshall et al., 2010). Finally, the French version of the STAQ asks a question on time spent in all forms of computer, tablet and video game use, and has shown to have good reliability (ICC = 0.64, 95% CI [0.38–0.80]) (Mensah et al., 2016).

Among adults, the Salmon SB questionnaire used three-day logs to validate self-reported computer use (r = 0.60) (Salmon et al., 2003). Only one study was found to compare a specific modality of SB with an appropriate objective measure. The Workplace Computer Use Questionnaire compared self-reported occupational computer use to direct observation and found they were moderately correlated (r = 0.41, p = 0.001); reliability was not assessed (Douwes, De Kraker & Blatter, 2007).

Total screen time

The ASAQ reported excellent reliability (grade 6 girls: ICC = 0.76, 95% CI [0.57–0.87] to grade 8 boys: ICC = 0.90, 95% CI [0.82–0.95]) for the measure of total screen time, which was calculated as the sum of all time watching TV, videos, DVDs, and using a computer for fun or homework (Hardy, Booth & Okely, 2007). The STAQ (ICC = 0.70, 95% CI [0.48–0.84]) (Mensah et al., 2016) and Domain-Specific Last 7-d Sedentary Time Questionnaire (SIT-Q-7d) (average day ICC = 0.61, 95% CI [0.53–0.67]) (Wijndaele et al., 2014) also demonstrated good reliability for total screen time calculated as the sum of individual screen-based behaviours in adults.

Reading

We were unable to identify any studies examining the reliability or validity of reading questions in children and youth. Although the ASAQ includes a question on reading, to our knowledge its reliability and validity have not been reported. In contrast, several questionnaires have undergone psychometric testing for items related to reading in adults. The Salmon SB questionnaire had the best level of reliability for reading with an ICC of 0.78 (95% CI [0.69–0.84]) (Salmon et al., 2003; Gardiner et al., 2011). The MOST (adapted from Salmon’s questionnaire; ICC = 0.74, 95% CI [0.51–0.86]) (Gardiner et al., 2011), SBQ (weekday: ICC = 0.64, 95% CI [0.44–0.78], weekend: ICC = 0.48, 95% CI [0.24–0.67]) (Rosenberg et al., 2010) and Sit-Q-7D (ICC = 0.59, 95% CI [0.51–0.66]) (Wijndaele et al., 2014) had slightly lower reliability, although it should be pointed out that there were only minor differences in wording across the three questionnaires, and all ICCs fell in the “fair to excellent” range. Reading time from the Salmon SB questionnaire was validated against a three-day log and a low correlation between the two measures (r = 0.20) was reported (Salmon et al., 2003).

Stationary transportation

The reliability of the Pre-PAQ proxy-report questionnaire ranged from poor to good (ICC = 0.31–0.63, 95% CI: NR) for a question focusing on the amount of car time over the past week in pre-school aged children (Dwyer et al., 2011). The ASAQ question focusing on time spent in a car, bus or train has good reliability (average ICC = 0.61) in boys and girls in grades 6, 8 and 10, but performed significantly better in girls than boys (e.g., grade 10 girls ICC = 0.93, 95% CI [0.85–0.97] vs. grade 10 boys: ICC = 0.25, 95% CI [−0.31–0.57]) (Hardy, Booth & Okely, 2007). Among adults, the International Physical Activity Questionnaire (IPAQ; r = 0.81 − 0.91) (Rosenberg et al., 2008) and the Salmon SB questionnaire (ICC = 0.85, 95% CI [0.79–0.89]) (Salmon et al., 2003) had excellent reliability for weekly passive transport. The SBQ also has excellent reliability for both weekday (ICC = 0.76, 95% CI [0.61–0.86]) and weekend days (ICC = 0.72, 95% CI [0.56–0.83]) (Rosenberg et al., 2010). The SIT-Q had good reliability for both weekday (ICC = 0.65, 95% CI [0.48–0.77]) and weekend days (ICC = 0.51 (95% CI [0.30–0.67]) (Lynch et al., 2014).

Total sedentary behaviour

Total SB was the only outcome for which we could find comparisons to appropriate objective standards in any age group. Among children and youth, estimated after-school SB (a composite score of TV, computer and cell-phone time) from the Youth Activity Profile (YAP) was highly correlated (r = 0.75, P < 0.001) with total sedentary time from the Sensewear armband (Saint-Maurice & Welk, 2015). The Activity Questionnaire for Adults and Adolescents (AQuAA; r = 0.23, P > 0.05), (Chinapaw et al., 2009) COMPASS (r = 0.20; p < 0.05) (Leatherdale, Laxer & Faulkner, 2014) and Physical Activity and Sedentary Behavior Assessment Questionnaire (PASBAQ) (r = 0.20 − 0.27) (Scholes et al., 2014) reported low correlations between self-reported total SB (calculated as the sum of all SB modalities) and hip-worn accelerometers in pediatric populations. Importantly, the COMPASS questionnaire also presented with high levels of test-retest reliability (ICC = 0.79, 95% CI: NR) (Leatherdale, Laxer & Faulkner, 2014). We did not identify any studies examining the validity of questions of total sitting time in children and youth, though most of the items for total SB are likely to be accomplished while sitting.

Among adults, validation studies have looked at single item estimates of sitting time, or have generated a composite score from a number of items to estimate total SB. The Past-day Adults’ Sedentary Time (PAST) and Past-Adults’ Sedentary Time - University (PAST-U) questionnaires had the highest measures of validity (PAST: r = 0.57, 95% CI [0.39–0.71], PAST-U: r = 0.63, 95% CI [0.44–0.76]) between a total of sum of SBs and sedentary time from the activPAL (Clark et al., 2013; Clark et al., 2016). The questionnaire from the AusDiab3 Study (r = 0.46, 95% CI [0.40–0.52]) (Clark et al., 2015) and the Madras Diabetes Research Foundation Physical Activity Questionnaire (MPAQ; r = 0.48, 95% CI [0.32–062]) (Anjana et al., 2015) also had moderate agreement with objective measures. In addition, the MPAQ also had excellent reliability for all sitting time (ICC = 0.81, 95% CI [0.78–0.84]) (Anjana et al., 2015). The Salmon SB questionnaire had excellent reliability for total SB (ICC = 0.79, 95% CI [0.71–0.85]) (Salmon et al., 2003). Even though the IPAQ is one of the most frequently used tools for self-reported SB, it relates poorly to objective measures. The validity of the IPAQ has been examined in multiple studies using accelerometers and inclinometers, with correlations generally ranging between 0.22 and 0.50 (depending on study sample), but with correlations for test-retest reliability generally above 0.70 (Rosenberg et al., 2008; Craig et al., 2003; Kolbe-Alexander et al., 2006; Umstattd Meyer et al., 2013).

Discussion

The purpose of the present review was to summarize the questions used to assess SB in national and international population surveillance surveys, and to identify the most valid and reliable questions for measuring both total SB and specific sub-domains and modes of SB. Although we identified a large number of national/international surveys, as well as a relatively large number of questionnaires with published results from psychometric testing, we found there was relatively little overlap between the two groups. Questions used in large population health surveys have typically not undergone appropriate evaluation with respect to validity or reliability, whereas questionnaires that have undergone this psychometric testing have typically not been used in larger national/international surveys.

Of the various modalities of SB, available evidence suggests that in general, self-reported total SB, TV viewing, computer use, and total screen time are negatively associated with physical and psychosocial health indicators in both children and adults (Carson et al., 2016; Ford & Caspersen, 2012; Grontved & Hu, 2011). Although it has been the focus of relatively few studies, the opposite relationship is observed for reading, which is associated with higher levels of academic achievement in children, and increased longevity in adults (Carson et al., 2016; Bavishi, Slade & Levy, 2016). It is unclear whether these relationships are due to physiological mechanisms, or due to confounding via other variables (e.g., socio-economic status), though at present there is little evidence to suggest that reading per se has a negative impact on health. Limited evidence suggests that transportation-related and occupational sedentary time may also be associated with poor health outcomes (Sugiyama et al., 2016; Van Uffelen et al., 2010). However, to our knowledge there is no evidence to suggest that the health impact of occupational sedentary time is different from that of total sedentary time, or that the impact of occupational computer use is different than that of non-occupational computer use. A sum of all modalities of SB is important for providing prevalence estimates of sedentary time; however, specific modalities of SB associate differently with health and are useful for surveillance. Given their consistent and deleterious associations with health indicators, and high prevalence of daily use, we suggest that TV time, computer time and total screen time are the self-report modalities of SB of greatest importance to include in population health surveys. We also suggest that if feasible, time spent in sedentary transport and reading are worth measuring and may provide insightful information.

As noted earlier, objective measurement tools (e.g., inclinometers and accelerometers) can only be used to test the validity of questions, or series of questions, aimed at estimating total sedentary time. The studies included in this review show poor validity in total SB when various questionnaires are assessed against objective measures. Similarly, Hidding et al. reported an absence of SB questionnaires that are both reliable and valid for use among children and youth (Hidding et al., 2016). Important to consider is that although accelerometers and inclinometers can help to validate sitting time questionnaires, they are unable to tell if a specific question accurately assesses specific modalities of SB (e.g., TV viewing, computer use, etc.). The questionnaires that performed best when compared to objective measures, specifically the PAST (Clark et al., 2013) and PAST-U (Clark et al., 2016), asked participants to record their time spent in nine different modes of SB, the sum of which provided a measure of total SB time. It is recognized, however, that a nine-item questionnaire is likely prohibitively long for inclusion in population surveillance surveys that are designed to obtain broad-level indicators of health across a large number of areas. The review was unable to locate any studies that examined the validity criterion of questions measuring screen time, reading or sedentary transportation. This is not surprising given the inability of objective measurement devices to delineate one type of SB from another. Thus, it is unclear whether answers to these questionnaires represent an accurate depiction of an individual’s time spent in highly prevalent modalities of SB. It is also important to identify the main limitation of this paper; the absence of a systematic and comprehensive search strategy. It is therefore likely that there are questionnaires/surveys that have not been captured in the review.

Importantly, while the validity of most self- and proxy-report SB health surveillance surveys are unknown, they still appear to provide useful measures of risk associated with health behaviours. In fact, self-reported SBs tend to be more strongly associated with health outcomes than objective measures, especially among children and youth (Carson et al., 2016; Stamatakis et al., 2012; Cliff et al., 2016). This suggests that it may be the behaviours done while sedentary (e.g., watching TV vs. reading) that are more important than total SB (Cliff et al., 2016). In addition, recall of specific SBs like screen time is likely easier than recalling all instances of sitting time throughout the day. Further, the available evidence does not suggest that SB questionnaires are invalid; rather that the validity of most questionnaires, especially those used in national/international surveillance surveys, have not been assessed against appropriate criterion measures. As noted elsewhere, objective and subjective measures of SB provide different, but complementary, information (Saunders, Chaput & Tremblay, 2014). Therefore, it is recommended that population health surveys consider employing both types of measures where feasible (i.e., both an inclinometer and a questionnaire).

In contrast to validity, we identified several questionnaires with acceptable reliability for the assessment of various SB domains in both adults and children. Reliability is a key factor for population surveillance surveys where the assessment of SBs over time are important to monitor the prevalence of this risk factor, as well as to evaluate changes resulting from population-level interventions (Herman & Saunders, 2016). While it would be ideal to have access to questionnaires that are known to be both valid and reliable, it is still useful and important to know that reliable options do exist for the measurement of important SB modalities. It is important to consider that a tool that has shown to be reliable at one time point, may lose its relevance and require updating with the emergence of new modes of SB as a result from changes in technology and its use. We recognize that reliability results did vary substantially between measures. Some of this variation may be a result of the population in which reliability of the questionnaire was assessed (e.g., general vs. special population) and the context (e.g., study looking only at reliability and validity of questionnaire vs. assessing reliability and validity within a pre-existing study).

Additional factors for consideration

In addition to validity and reliability, there are several other factors of relevance when attempting to determine the ideal means for assessing SB in population surveillance surveys. For example, it has been noted that individuals are increasingly engaged in “multi-tasking”, whereby they are participating in multiple forms of SB simultaneously (Rideout, Foehr & Roberts, 2010). For example, individuals may be reading or playing a video game on a tablet while also watching TV. If the total time spent doing each of these activities is simply summed together, this can result in inflated estimates for total screen time or total SB (Saunders, Prince & Tremblay, 2011). Some of the questionnaires identified in this review (e.g., the MOST questionnaire) address this issue using a pre-amble to ask respondents to only identify the “main” form of SB during a given time period. We recommend that future surveys incorporate this methodology.

It is important that surveillance surveys assess the types of SBs that reflect those which are most used in the population and recognize that these may change over time. For example, many individuals now watch television programming over internet streaming services such as Netflix or YouTube in addition to (or instead of) traditional cable or satellite TV. One option to ensure that the most current SB modes are assessed is through the consistent use of relatively generic questions for each SB modality, with detailed examples provided beneath that can be updated as new forms of SB emerge.

We also recommend that population surveillance surveys ensure that the questions used to measure SB are in a format that can assess whether the population is meeting relevant public health guidelines. For example, Canadian guidelines recommend that school-aged children and youth accrue no more than two hours per day of recreational screen time (Tremblay et al., 2016). It is therefore important that population health surveys provide information in a format which can be used to assess whether or not an individual is meeting such guidelines. In particular, allowing respondents to enter their response as a specific continuous number (e.g., hours and minutes per day or week), or providing a large range of individual options (e.g., 0, 30 min, 1 h, 2 hour…6+ hours) allows this to be easily calculated. These approaches have been used by several questionnaires with high levels of reliability in both children and adults (e.g., COMPASS, SBQ). Importantly however, scaled response categories preclude the ability to determine specific durations of SB for those in the highest category (e.g., if a person answers “6+ hours” you will not know if they engaged in 6 h versus 12 h of screen time). This can complicate data analysis, as well as in the interpretation of “average” time spent in a SB across a population.

Finally, with respect to population surveillance surveys, it is important that they remain consistent, whenever possible, to provide information on secular trends in SBs over time. The questions used to assess SB vary widely across national and international surveys, often change over time, and do not always target the same domains of SB (e.g., screen time, leisure, occupational, transport, etc.). These issues preclude meaningful comparisons over time, or across countries and regions, and diminish the usefulness of the information provided by these surveys/questionnaires for researchers, health behaviour interventionists and policy makers. Thus, it is recommended that population health surveys should use consistent questions from year-to-year whenever possible.

Suggested SB module

Table 1 provides a suggested SB module that we developed using modified individual questions from other questionnaires with acceptable reliability. Examples have been provided in brackets for some questions; these can be updated over time as new popular modes of SB emerge (e.g., a new Smartphone or internet streaming service). We have proposed individual questions for time spent using screens, watching TV, using computers (including tablets, smart phones, and video games), reading, in sedentary transportation, and total sitting time. To address SB guidelines for children and youth, the caveat “during your free time” can be added for questions related to screen time for children and youth, but not adults (Tremblay et al., 2016). For each question, answers are reported in a continuous fashion using hours and minutes per week. This approach allows the researcher to easily determine whether an individual is meeting or exceeding public health guidelines, which can be difficult (and sometimes impossible) when using categorical variables. As noted above, this is the approach used by several questionnaires that performed well on test-retest reliability. The preambles from the MOST and SBQ questionnaires have also been adapted in an attempt to minimize the impact of multitasking.

Table 1 Suggested sedentary behaviour module in English and French.

ISAT (International Sedentary Assessment Tool).	
Outil international d’évaluation du comportement sédentaire (ISAT—Version française)	
The following questions are about activities you/your child did over the past week while sitting, reclining or lying down. Do not count the time you/they spent in bed sleeping or napping.	
Les questions suivantes portent sur des activités que vous (ou votre enfant) avez réalisées durant la dernière semaine, alors que vous étiez assis ou allongés. Ne pas considérer le temps que vous (ou votre enfant) avez passé au lit à dormir ou faire la sieste.	
For each of the following activities (question 1, 2, 3, and 6) only count the time when this was your/their main activity.	
For example if you/they are watching television and surfing the internet, count it as television time or computer time, but not as both. [adapted from MOST questionnaire]	
Ne comptez le temps alloué à chacune des activités suivantes (question 1, 2, 3 et 6) que lorsqu’il s’agissait de votre activité principale (ou de celle de votre enfant). Par exemple, si vous (ou votre enfant) regardez la télévision ET naviguez sur Internet, veuillez compter soit le temps de télévision soit le temps d’ordinateur, mais non les deux.	
On a typical WEEKDAY/WEEKEND DAY in the past week, how much time do you/your child spend sitting, reclining or lying down and…[adapted from SBQ and MOST questionnaires]	
La semaine passée, lors d’une journée habituelle de semaine/fin de semaine, combien de temps avez-vous (ou votre enfant) passé assis ou allongé à…	
SEDENTARY ITEM
ACTIVITÉS SÉDENTAIRES	TIME
TEMPS ALLOUÉ	SOURCE
SOURCE	MODIFICATIONS
MODIFICATIONS	
1. Watching TV or using a computer, tablet or smartphone or [for children and youth only: during your/their free time?]*
1. Regarder la télévision ou utiliser votre ordinateur, tablette ou téléphone intelligent [pour les enfants et adolescents seulement: lors de ton/son temps libre?]*	__ hours
__ heures	_____ minutes
_____ minutes	CHMS
ECMS	iPad is no longer specifically referenced in question.
L’utilisation d’un iPad n’est plus spécifiée dans cette question	
(Count time watching videos, playing computer games, emailing or using the Internet. Do not include time spent on a computer at work or at school.)
(Compter le temps passé à regarder des vidéos, jouer sur l’ordinateur, consulter ses courriels ou naviguer sur Internet. Ne pas inclure le temps passé sur un ordinateur au travail ou à l’école.)
*Note: this question can be omitted if question 2 & 3 are used instead.
*Note: cette question peut ne pas être utilisée si les questions 2 & 3 sont utilisées à la place.					
2. Watching television or videos [for children and youth only: during your/their free time?]
2. Regarder la télévision ou des vidéos [pour les enfants et adolescents seulement: lors de ton/son temps libre?]*	__ hours
__ heures	_____ minutes
_____ minutes	MOST	Addition of “during your free time”, and information in parentheses.
Ajout de “pendant votre temps libre”, et information entre parenthèses.	
(Count time spent watching television, DVDs and online videos)
(Compter le temps passé à regarder la télévision, des DVD et des vidéos en ligne)					
3. Using a computer [for children and youth only: during your/their free time?]
3. Utiliser un ordinateur [pour les enfants et adolescents seulement: lors de ton/son temps libre?]*	__ hours
__ heures	_____ minutes
_____ minutes	MOST	Added “during free time”, removed “internet” from main question, placed examples in parentheses.
Ajout de “pendant votre temps libre”, suppression de “internet” de la question principale, exemples placés entre parenthèses.	
(Count time spent on things such as computers, laptops, Xbox, PlayStation, iPod, iPad or other tablet, or a smartphone, YouTube, Facebook or other social networking tools, and the Internet).
(Compter le temps passé à utiliser un ordinateur, un ordinateur portable, une console de jeux vidéo comme Xbox ou PlayStation, un iPod, un iPad ou toute autre tablette, un téléphone intelligent, YouTube, Facebook ou autre réseau social et Internet).					
4. During the last 7 days, how much time did you usually spend sitting on a week/weekend day?
4. De façon générale, au cours des 7 derniers jours, combien de temps avez-vous passé assis lors des jours de semaine et fin de semaine?	__ hours
__ heures	_____ minutes
_____ minutes	IPAQ
IPAQ	Information from preamble moved to parentheses.
Ajout entre parenthèses des renseignements du préambule.	
(Include time spent at school or work, at home, while doing course work, and during leisure time. This may include time spent sitting at a desk, visiting friends, reading or sitting or lying down to watch television).
(Inclure le temps passé à l’école, au travail, à la maison, à faire les devoirs et pendant les loisirs. Cela peut inclure le temps passé assis à un bureau, avec des amis ou assis ou allongé à lire ou regarder la télévision.)					
5. Sitting and driving in a car, bus, or train?
5. Conduire une voiture ou à être dans l’autobus ou le train ?	__ hours
__ heures	_____ minutes
_____ minutes	SBQ
SBQ	N/A
N/A	
6. Sitting reading a book or magazine?
6. Lire un livre ou un magazine?	__ hours
__ heures	_____ minutes
_____ minutes	SBQ and CHMS
SBQ et CHMS	N/A
N/A	
(Only include reading during your free time. Include reading done using electronic formats. Include time spent reading as part of your homework, but do not include time spent reading at work, during class time or while exercising).
(Seulment inclure la lecture pendant votre temps libre. Inclure la lecture sur un appareil électronique et le temps passé à lire pour les devoirs d’école. Ne pas inclure, le temps passé à lire au travail, à l’école ou alors que vous faisiez de l’activité physique.)					
Notes.

Information in square brackets is provided for the reader, but should not be included on the final questionnaire.

Les renseignements entre crochets ne doivent pas être inclus dans la version finale du questionnaire.

Reading was included given that it is the only form of SB consistently associated with positive health indicators (Bavishi, Slade & Levy, 2016). At present it is unclear whether the health impacts of reading on a screen-based device differ from those of reading a physical book. Studies that have shown associations between reading and academic achievement or longevity tend to simply ask how much time people spend readingbooks or magazines, without specifying the device used (Bavishi, Slade & Levy, 2016; Romer, Bagdasarov & More, 2013). As books and magazines are likely to be increasingly read on screen-based devices, more research will be needed to determine if this has any impact on the relationship between reading and health, which may also differ based on the specific screen-based device being use (e.g., lit screens may have a more detrimental impact on sleep than non-lit screens (Hale & Guan, 2015)). For now, it is suggested to include wording similar to that used in the 2015 CHMS (Statistics Canada, 2016), which includes reading done using both physical books and electronic devices.

The questions are listed in order of their importance, based on their associations with health outcomes. The options also recognize the need for population surveys that include SB measures may have limited space for questions regarding a single health behaviour. Therefore, if there is room for only one question, then question 1 (Screen time) should be used. If there is room for two questions, then question 2 (TV time) and 3 (Computer time) should be used; this allows the researcher to also calculate total screen time (i.e., will provide a response for question 1). If the survey allows for more items, we suggest adding question 4 to 6 sequentially.

To date, many SB questionnaires have separated weekdays from weekend days. This is especially true in the pediatric population, where the majority of questionnaires separate week (or school) and weekend days. This format is recommended as individuals often have very different and sometimes counter-intuitive schedules on weekdays versus weekend days. In line with this practice, we have suggested that each question be asked twice; once for weekdays, and once for weekend days.

Conclusions

This review aimed to describe SB modules that have been commonly used in national and international surveys. We also aimed to identify the most reliable and valid tools currently available to assess SB. Unfortunately, we were unable to identify a single tool that met all of our criteria. As such, we have recommended a new module, based on the best available evidence that can be modified to suit the needs of individual surveys. Future research could investigate the psychometric properties of the proposed module, as well as other questionnaires currently used in national and international population health surveys.

Supplemental Information

Table S1 Sample MEDLINE search strategy

Click here for additional data file.

Table S2 Sedentary behaviour questions from national population health surveys

CI, confidence interval; cpm, count per minute; ICC, intraclass correlation coefficient; LoA, limits of agreement; NS, not significant; UK, United Kingdom; USA, United States of America; WHO, World Health Organization.

Click here for additional data file.

Table S3 Sedentary behaviour questionnaires with psychometric testing

*, Total males and females, separate data not shown; %F, percentage of sample that is female; cpm, counts per minute; ICC, intraclass correlation coefficient; LoA, limits of agreement; N/A, not applicable; NR, not reported; NS, not significant; r, correlation coefficient; SB, sedentary behaviour; SD, standard deviation; UK, United Kingdom; USA, United States of America.

Click here for additional data file.

The authors would like to thank Dr. Jean-Philippe Chaput, Dr. David Thivel and Dr. Geneviève Leduc for the French translation of the International Sedentary Assessment Tool. Stephanie Prince is a Canadian Institutes for Health Research (CIHR)-Public Health Agency of Canada Health Systems Impact Fellow and Allana LeBlanc is a CIHR-Ottawa Model for Smoking Cessation Health Systems Impact Fellow. Travis Saunders is the senior author of the publication.

Additional Information and Declarations

Competing Interests

Author Contributions

Data Availability

Stephanie Prince has received an equipment competition award from PAL Technologies Ltd. Travis Saunders has received research support from Stepscount Inc, and in-kind support from Stepscount Inc, Ergotron and Fitabase. Allana LeBlanc and Rachel Colley declare that they have no conflicts of interest. The Public Health Agency of Canada provided feedback on the original draft of the manuscript. The views expressed in this paper are solely those of the authors and do not reflect those of Statistics Canada.

Stephanie A. Prince and Allana G. LeBlanc performed the experiments, analyzed the data, contributed reagents/materials/analysis tools, wrote the paper, prepared figures and/or tables, reviewed drafts of the paper.

Rachel C. Colley analyzed the data, contributed reagents/materials/analysis tools, reviewed drafts of the paper.

Travis J. Saunders conceived and designed the experiments, performed the experiments, analyzed the data, contributed reagents/materials/analysis tools, wrote the paper, prepared figures and/or tables, reviewed drafts of the paper.

The following information was supplied regarding data availability:

The research in this article did not generate any data or code; this article is a literature review.

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
