# Peer review of "Measurement of sedentary behaviour in population health surveys: a review and recommendations"

_PeerJ, doi:10.7717/peerj.4130_

## Round 0.1 · original submission · Major Revisions

Please take careful note of the reviewer's comments and suggested revisions and, please provide a point by point response to each issue raised and how/where in the revised manuscript the issue was addressed.

Reviewer 1 ·

Basic reporting

I would like to thank the authors of this manuscript for a very interesting study. The topic of sedentary behaviour in population health surveys is very important and it is very suitable for the scope of this journal. Sufficient background and context was provided. However, more information can be added in several paragraphs to make the manuscript more clear for the readers.
Line 67) Provide citations
Line 189) "Figure 1 provides results of the search strategy used to identify 35 papers that examined the validity"… This is not very clear. Do you mean Figure 1 provides results of the search strategy used to identify [37?] papers that examined the validity?

In addition, figure 1 can be used to breakdown the number of included papers undergone psychometric testing in a pediatric population, and in the adult population.

For figure 1, it is recommended to provide reasons for excluding articles at full-text screening and the number of articles that are excluded for each reason. Please see “Item 17: Study selection Give numbers of studies screened, assessed for eligibility, and included in the review, with reasons for exclusions at each stage, ideally with a flow diagram http://www.bmj.com/content/bmj/339/bmj.b2700.full.pdf “ . Since this study is not a systematic review, this is not a serious issue. I just wanted to mention it.

In Supplemental_Table_2
Line 184) The reliability and validity of individual questionnaires (n=34 or 37?) has been summarized in Supplemental table 2. Also, I found it to be a bit confusing to know which papers have been discussed in the previous systematic reviews [16, 17] and which ones you are investigating in this study. It would be great if you could clarify.

Supplemental table 3 is not reference anywhere in the manuscript.

Experimental design

In order to conduct a comprehensive search it is very important to include mesh terms in the search. It is impossible to come up with a comprehensive list of search terms. Mesh searching will assist with retrieving all potential articles relevant to a topic. For example, as a Subjective measurement term, one might use self-rated instead of self report. By searching for "Self Report"[Mesh], you can reduce the risk of missing that specific article. I strongly suggest adding the following mesh terms: "reproducibility of results"/ OR validation studies as topic/ Validation Studies/

Mesh searching becomes even more important if you limit the search to tw field.
"The Text Word (TW) index is an alias for all of the fields in a database which contain text words and which are appropriate for a subject search. The Text word index in Ovid MEDLINE (R) includes Title (TI) and Abstract (AB)". (from http://ospguides.ovid.com/OSPguides/medline.htm#TW ) basically tw exclude searching the author's keywords field and mesh field.

One of the main limitations of this study is the lack of a comprehensive search. I think it would be great to mention this in the article to increase transparency.

Line 123) I do not think searching ‘EBM Reviews - Cochrane Database of Systematic Reviews’ would have been useful as the authors were aiming to find original research articles rather than systematic review articles.

Validity of the findings

As mentioned before one of the main limitations of this study is the lack of a comprehensive search. I think it would be great to mention this in the article to increase transparency.

Reviewer 2 ·

Basic reporting

Overall a well conducted review. Some stylistic suggestions are made in section 4 General comments.

Experimental design

Methods are appropriate.

Given there were two recently published systematic reviews that summarized the psychometric properties of sedentary behaviour questionnaires in children and youth, I suggest the authors restrict their paper to adult questionnaires. This would make the tables and text somewhat more manageable. There is little benefit to generating a third systematic review on the same topic (so soon as the first two).

Validity of the findings

No meta-analysis or statistical synthesis applied. Narrative description of surveys and their psychometric properties only.

The narrative description of results provided in the text is too vague. Statements such as ‘Computer and/or video game time were also frequently assessed’ (line170) should be quantified by including the number of studies and percentage. Another example is ‘Many, but not all, of the surveys’ (line 173).

Despite the authors stating that they included questionnaires listed on the Sedentary Behaviour Questionnaire list compiled by the SBRN, there appear to be a number of questionnaires missing, e.g. the Bouchard Physical Activity Questionnaire, the SIT-Q. The Bouchard Physical Activity Questionnaire link from the SBRN site goes to an Am J Clin Nutr article behind a paywall, so this could not be further interrogated easily. The SIT-Q validation paper in BMC Public Health appears to meet the criteria for this review, so it’s not clear why this questionnaire was excluded.

Comments for the author

Introduction
lines 63-65 the authors suggest that evidence supports the idea that different types of sedentary behaviours have different health effects. This is not impossible, but differences in associations with health outcomes are probably a reflection of differential measurement error and/or confounding, rather than different physiological adaptations to sedentary behaviour in different settings.

line 75 providing readers with information to support future survey development. Given the plethora of surveys already in existence, do we really want more? Perhaps it would be better to suggest that better psychometric testing and refinement of existing measures be undertaken.

Methods
line 118 suggest removing the word “Unfortunately” from beginning of sentence.

Results
Throughout the text the ICCs are reported in a non-uniform fashion. Ideally, the ICC and 95% CIs should be given. In some places the authors simply give the ICC (eg 228), in others it appears to be the CIs only (eg line 242).

The first sentence of the paragraph beginning on line 284 is not well worded. I think the authors mean to say something like “validation studies have looked at both single-item estimates of sitting time, or have generated a composite score from a number of items to estimate total sedentary behavior”.

Discussion
Again, on line 313 the authors note that there seem to be positive health associations for reading (compared to the negative associations seen for other sedentary behaviours). Be very careful with the description and interpretation of this – likely to be confounding, not an actual different physiological adaptation.

Lines 325 – 326: this sentence is unfinished.

Line 341: actually, a number of questionnaires assessed the validity of screen time, reading and sedentary transportation (SIT-Q; SIT-Q-7d). Do the authors mean “examined the criterion validity”?

Line 351: confounding concept again.

---

## Round 0.2 · Minor Revisions

The revised manuscript is much improved; however, there are still a few outstanding issues that need more fully addressed. Please provide a point by point response detailing how and where each issue was addressed in the re-revised manuscript.

Reviewer 1 ·

Basic reporting

I have no further suggestions.

Experimental design

I have no further suggestions.

Validity of the findings

I have no further suggestions.

Comments for the author

Thank you for responding to all my comments. I have no further suggestions.

Reviewer 2 ·

Basic reporting

The authors have taken the reviewers' comments on board and done a decent job in revising their work. The manuscript is generally well written.

Experimental design

No comment

Validity of the findings

Original comment: Despite the authors stating that they included questionnaires listed on the Sedentary Behaviour Questionnaire list compiled by the SBRN, there appear to be a number of questionnaires missing, e.g. the Bouchard Physical Activity Questionnaire, the SIT-Q. The Bouchard Physical Activity Questionnaire link from the SBRN site goes to an Am J Clin Nutr article behind a paywall, so this could not be further interrogated easily. The SIT-Q validation paper in BMC Public Health appears to meet the criteria for this review, so it’s not clear why this questionnaire was excluded.

Revised comment: The authors have given a satisfactory response in relation to the Bouchard PAQ, however, they do not seem to appreciate that the SIT-Q and SIT-Q-7d are different questionnaires. The SIT-Q is (the original) domain-specific questionnaire with a past year recall period. It's development and psychometric properties are published here:

Lynch BM, Friedenreich CM, Khandwala F et al. BMC Public Health. 2014 Sep 1;14:899.

The SIT-Q-7d is a revised version that has changed the recall timeframe to 7 days. This version also removed some of the original items and added others (specifically relating to snacking):

Wijndaele K, DE Bourdeaudhuij, Godino JG et al. Med Sci Sports Exerc. 2014 Jun;46(6):1248-60.

Both questionnaires should be included in this review.

Comments for the author

Original comment: Throughout the text the ICCs are reported in a non-uniform fashion. Ideally, the ICC and 95% CIs should be given. In some places the authors simply give the ICC (eg 228), in others it appears to be the CIs only (eg line 242).

Revised comment: Restricting the reported findings to ICC only is problematic. The reader cannot draw meaningful conclusions or make an informed evaluation about the psychometric properties of the questionnaires based on ICC alone. I suggest the authors contact the researchers who have published their validation studies without 95% CIs and ask for these data to be supplied for the review paper. After doing this, please include 95% CIs where these are available, and in the case of non-response from original authors please indicated that 95% CIs are not available in your review paper (e.g. by writing something like "for TV viewing time the XXX questionnaire demonstrated reasonable convergent validity ICC=0.68, 95% CI not reported."

---

## Round 0.3 · Minor Revisions

Please could you address these remaining minor comments from the reviewer, before we can accept this paper for publication.

Reviewer 2 ·

Basic reporting

N/A

Experimental design

N/A

Validity of the findings

N/A

Comments for the author

Thank you for revising the manuscript and including the CIs for ICCs where these were available. This strengthens the reporting and will ensure readers utilising this review correctly cite the reliability and validity outcomes for the measures reported.

Thank you also for including the adult questionnaire previously omitted (the SIT-Q) in the Table. However, I note that none of the SIT-Q findings have been incorporated into the text. I think it is important to do so, as this questionnaire demonstrates very good ICC for TV (0.84, 95% CI: 0.75, 0.90) but poor computer ICC (0.31, 95% CI: 0.07, 0.52). It is also one of the few measures that reports sitting during transport.

---

## Round 0.4 · accepted · Accept

Thank you for making these final minor changes. I am happy to accept your manuscript for publications now. Congratulations.